# Sequence Modeling with Unconstrained Generation Order

**Dmitrii Emelianenko**[1,2]    **Elena Voita**[1,3]    **Pavel Serdyukov**[1]

[1]Yandex, Russia
[2]National Research University Higher School of Economics, Russia
[3]University of Amsterdam, Netherlands
{dimdi-y, lena-voita, pavser}@yandex-team.ru

## Abstract

The dominant approach to sequence generation is to produce a sequence in some predefined order, e.g. left to right. In contrast, we propose a more general model that can generate the output sequence by inserting tokens in any arbitrary order. Our model learns decoding order as a result of its training procedure. Our experiments show that this model is superior to fixed order models on a number of sequence generation tasks, such as Machine Translation, Image-to-LaTeX and Image Captioning.[1]

## 1   Introduction

Neural approaches to sequence generation have seen a variety of applications such as language modeling [1], machine translation [2, 3], music generation [4] and image captioning [5]. All these tasks involve modeling a probability distribution over sequences of some kind of tokens.

Usually, sequences are generated in the left-to-right manner, by iteratively adding tokens to the end of an unfinished sequence. Although this approach is widely used due to its simplicity, such decoding restricts the generation process. Generating sequences in the left-to-right manner reduces output diversity [6] and could be unsuited for the target sequence structure [7]. To alleviate this issue, previous studies suggested exploiting prior knowledge about the task (e.g. the semantic roles of words in a natural language sentence or the concept of language branching) to select the preferable generation order [6, 7, 8]. However, these approaches are still limited by predefined generation order, which is the same for all input instances.

Figure 1: Examples of different decoding orders: left-to-right, alternative and right-to-left orders respectively. Each line represents one decoding step.

In this work, we propose *INTRUS: INsertion TRansformer for Unconstrained order Sequence modeling*. Our model has no predefined order constraint and generates sequences by iteratively adding tokens to a subsequence in any order, not necessarily in the order they appear in the final

sequence. It learns to find convenient generation order as a by-product of its training procedure without any reliance on prior knowledge about the task it is solving.

Our key contributions are as follows:

- We propose a neural sequence model that can generate the output sequence by inserting tokens in any arbitrary order;

- The proposed model outperforms fixed-order baselines on several tasks, including Machine Translation, Image-to-LaTeX and Image Captioning;

- We analyze learned generation orders and find that the model has a preference towards producing "easy" words at the beginning and leaving more complicated choices for later.

## 2 Method

We consider the task of generating a sequence $Y$ consisting of tokens $y_t$ given some input $X$. In order to remove the predefined generation order constraint, we need to reformulate the probability of target sequence in terms of token insertions. Unlike traditional models, there are multiple valid insertions at each step. This formulation is closely related to the existing framework of generating unordered sets, which we briefly describe in Section 2.1. In Section 2.2, we introduce our approach.

### 2.1 Generating unordered sets

In the context of unordered set generation, Vinyals et al. [9] proposed a method to learn sequence order from data jointly with the model. The resulting model samples a permutation $\pi(t)$ of the target sequence and then scores the permuted sequence with a neural probabilistic model:

$$P(Y_\pi|x, \theta) = \prod_t p(y_{\pi(t)}|X, y_{\pi(0)}, .., y_{\pi(t-1)}, \theta). \tag{1}$$

The training is performed by maximizing the data log-likelihood over both model parameters $\theta$ and target permutation $\pi(t)$:

$$\theta^* = \arg\max_\theta \sum_{X,Y} \max_\pi \log P(Y_\pi|x, \theta). \tag{2}$$

Exact maximization over $\pi(t)$ requires $O(|Y|!)$ operations, therefore it is infeasible in practice. Instead, the authors propose using greedy or beam search. The resulting procedure resembles the Expectation Maximization algorithm:

1. E step: find optimal $\pi(t)$ for $Y$ under current $\theta$ with inexact search,
2. M step: update parameters $\theta$ with gradient descent under $\pi(t)$ found on the E step.

EM algorithms are known to easily get stuck in local optima. To mitigate this issue, the authors sample permutations proportionally to $p(y_{\pi(t)}|x, y_{\pi(0)}, .., y_{\pi(t-1)}, \theta)$ instead of maximizing over $\pi$.

### 2.2 Our approach

The task now is to build a probabilistic model over sequences $\tau = (\tau_0, \tau_1, ..., \tau_T)$ of insertion operations. This can be viewed as an extension of the approach described in the previous section, which operates on ordered sequences instead of unordered sets. At step $t$, the model generates either a pair $\tau_t = (pos_t, token_t)$ consisting of a position $pos_t$ in the produced so far sub-sequence ($pos_t \in [0, t]$) and a token $token_t$ to be inserted at this position, or a special EOS element indicating that the generation process is terminated. It estimates the conditional probability of a new insertion $\tau_t$ given $X$ and a partial output $\tilde{Y}(\tau_{0:t-1})$ constructed from the previous inserts:

$$p(\tau|X, \theta) = \prod_t p(\tau_t|X, \tilde{Y}(\tau_{0:t-1}), \theta). \tag{3}$$

**Training objective**   We train the model by maximizing the log-likelihood of the reference sequence $Y$ given the source $X$, summed over the data set $D$:

$$L = \sum_{\{X,Y\}\in D} \log p(Y|X,\theta) = \sum_{\{X,Y\}\in D} \log \sum_{\tau \in T^*(Y)} p(\tau|X,\theta) =$$
$$= \sum_{\{X,Y\}\in D} \log \sum_{\tau \in T^*(Y)} \prod_t p(\tau_t|X,\tilde{Y}(\tau_{0:t-1}),\theta), \tag{4}$$

where $T^*(Y)$ denotes the set of all trajectories leading to $Y$ (see Figure 2).

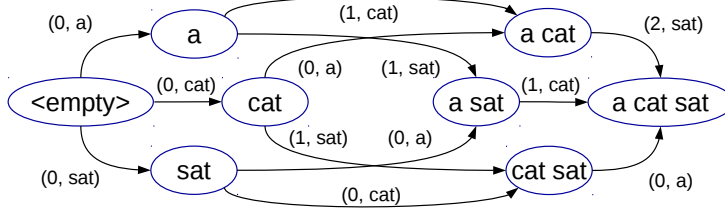

Figure 2: Graph of trajectories for $T^*(Y = $ "a cat sat").

Intuitively, we maximize the total probability "flowing" through the acyclic graph defined by $T^*(Y)$. This graph has approximately $O(|Y|!)$ paths from an empty sequence to the target sequence $Y$. Therefore, directly maximizing (4) is impractical. Our solution, inspired by [9], is to assume that for any input $X$ there is a trajectory $\tau^*$ that is the most convenient for the model. We want the model to concentrate the probability mass on this single trajectory. This can be formulated as a lower bound of the objective (4):

$$L = \sum_{\{X,Y\}\in D} \log p(Y|X,\theta) = \sum_{\{X,Y\}\in D} \log \sum_{\tau \in T^*(Y)} \prod_t p(\tau_t|X,\tilde{Y}(\tau_{0:t-1}),\theta) \geq$$
$$\geq \sum_{\{X,Y\}\in D} \log \max_\tau \prod_t p(\tau_t|X,\tilde{Y}(\tau_{0:t-1}),\theta) = \sum_{\{X,Y\}\in D} \max_\tau \sum_t \log p(\tau_t|X,\tilde{Y}(\tau_{0:t-1}),\theta). \tag{5}$$

The lower bound is tight iff the entire probability mass in $T^*$ is concentrated along a single trajectory. This leads to a convenient property: maximizing (5) forces the model to choose a certain "optimal" sequence of insertions $\tau^* = \arg\max_\tau \prod_t p(\tau_t|X,\tilde{Y}(\tau_{0:t-1}),\theta)$ and concentrate most of the probability mass there.

The bound (5) depends only on the most probable trajectory $\tau^*$, thus is difficult to optimize directly. This may result in convergence to a local maximum. Similar to [9], we replace max with an expectation w.r.t. trajectories sampled from $T^*$. We sample from the probability distribution over the trajectories obtained from the model. The new lower bound is:

$$\sum_{\{X,Y\}\in D} E_{\tau \sim p(\tau|X,\tau\in T^*(Y),\theta)} \sum_t \log p(\tau_t|X,\tilde{Y}(\tau_{0:t-1}),\theta). \tag{6}$$

The sampled lower bound in (6) is less or equal to (5). However, if the entire probability mass is concentrated on a single trajectory, both lower bounds are tight. Thus, when maximizing (6), we also expect most of the probability mass to be concentrated on one or a few "best" trajectories.

**Training procedure**   We train our model using stochastic gradient ascent of (6). For each pair $\{X,Y\}$ from the current mini-batch, we sample the trajectory $\tau$ from the model: $\tau \sim p(\tau|X, \tau \in T^*(Y), \theta)$. We constrain sampling only to correct trajectories by allowing only the correct insertion operations (i.e. the ones that lead to producing $Y$). At each step along the sampled trajectory $\tau$, we maximize $\log p(ref(Y,\tau_{0:t-1})|X,\tilde{Y}(\tau_{0:t-1}),\theta)$, where $ref(Y,\tau_{0:t-1})$ defines a set of all insertions $\tau_t$ immediately after $\tau_{0:t-1}$, such that the trajectory $\tau_{0:t-1}$ extended with $\tau_t$ is correct: $\tau_{0:t-1} \oplus \tau_t \in T^*(Y)$. From a formal standpoint, this is a probability of picking *any* insertion that is on the path to $Y$. The simplified training procedure is given in Algorithm 1.

**Algorithm 1: Training procedure (simplified)**

**Inputs:** batch $\{X, Y\}$, parameters $\theta$, learning rate $\alpha$, $\vec{g} := \vec{0}$      // $\vec{g}$ is the gradient accumulator
**for** $X_i, Y_i \in \{X, Y\}$ **do**
   $\tau \sim p(\tau | X_i, \tau \in T^*(Y_i), \theta)$
   **for** $t \in 0, 1, \ldots, |\tau| - 1$ **do**
      $ref := ref(Y, \tau_{0:t-1})$      // correct inserts
      $L_{i,t} = \log p(ref | X_i, \tilde{Y}_i(\tau^*_{0:t-1}), \theta)$
      $\vec{g} := \vec{g} + \frac{\partial L_{i,t}}{\partial \theta}$
   **end**
**end**
**return** $\theta + \alpha \cdot \vec{g}$

The training procedure is split into two steps: (i) pretraining with uniform samples from the set of feasible trajectories $T^*(Y)$, and (ii) training on samples from our model's probability distribution over $T^*(Y)$ till convergence. We discuss the importance of the pretraining step in Section 5.2.

**Inference**    To find the most likely output sequence according to our model, we have to compute the probability distribution over target sequences as follows:

$$p(Y|X, \theta) = \sum_{\tau \in T^*(Y)} p(\tau | x, \theta). \tag{7}$$

Computing such probability exactly requires summation over up to $O(|Y|!)$ trajectories, which is infeasible in practice. However, due to the nature of our optimization algorithm (explicitly maximizing the lower bound $E_{\tau \sim p(\tau | X, \tau \in T^*(Y), \theta)} p(\tau | x, \theta) \leq \max_{\tau \in T^*(Y)} p(\tau | x, \theta) \leq P(Y|X))$, we expect most of the probability mass to be concentrated on one or a few "best" trajectories:

$$P(Y|X) \approx \max_{\tau \in T^*(Y)} p(\tau | x, \theta). \tag{8}$$

Hence, we perform approximate inference by finding the most likely trajectory of insertions, disregarding the fact that several trajectories may lead to the same $Y$.[2] The resulting inference problem is defined as:

$$Y^* = \arg\max_{Y(\tau)} \log p(\tau | X, \theta). \tag{9}$$

This problem is combinatoric in nature, but it can be solved approximately using beam search. In the case of our model, beam search compares partial output sequences and extends them by selecting the $k$ best token insertions. Our model also inherits a common problem of the left-to-right machine translation: it tends to stop too early and produce output sequences that are shorter than the reference. To alleviate this effect, we divide hypotheses' log-probabilities by their length. This has already been used in previous works [10, 11, 12].

## 3   Model architecture

INTRUS follows the encoder-decoder framework. Specifically, the model is based on the Transformer [10] architecture (Figure 3) due to its state-of-the-art performance on a wide range of tasks [13, 14, 15]. There are two key differences of INTRUS from the left-to-right sequence models. Firstly, our model's decoder does not require the attention mask preventing attention to subsequent positions. Decoder self-attention is re-applied at each decoding step because the positional encodings of most tokens change when inserting a new one into an incomplete subsequence of $Y$.[3] Secondly, the decoder predicts the joint probability of a token and a position corresponding to a single insertion (rather than the probability of a token, as usually done in the standard setting). Consequently, the

predicted probabilities should add up to 1 *over all positions and tokens* at each step. We achieve this by decomposing the insertion probability into the probabilities of a token and a position:

$$p(\tau_t) = p(token|pos) \cdot p(pos),$$
$$p(pos) = softmax(H \times w_{loc}), \qquad (10)$$
$$p(token|pos) = softmax(h_{pos} \times W_{tok}).$$

Here $h_{pos} \in \mathbb{R}^d$ denotes a single decoder hidden state (of size $d$) corresponding to an insertion at position $pos$; $H \in \mathbb{R}^{t \times d}$ represents a matrix of all such states. $W_{tok} \in \mathbb{R}^{d \times v}$ is a learned weight matrix that predicts token probabilities and $w_{loc} \in \mathbb{R}^d$ is a learned vector of weights used to predict positions. In other words, the hidden state of each token in the current sub-sequence defines (i) the probability that the next token will be generated at the position immediately preceding current and (ii) the probability for each particular token to be generated next.

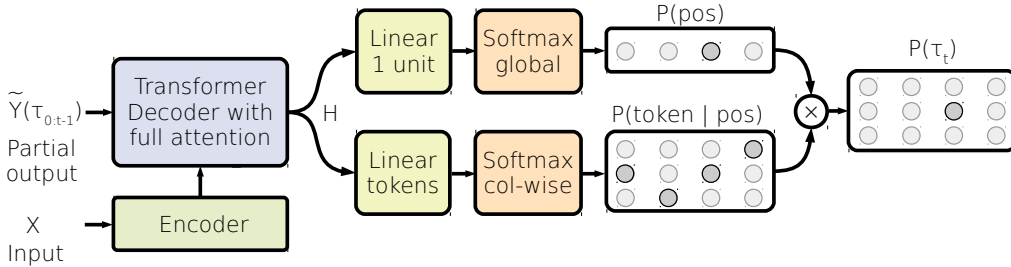

Figure 3: Model architecture: $p(\tau_t|X, \tilde{Y}(\tau_{0:t-1}), \theta)$ for a single token insertion. Output of the column $i$ for the token $j$ defines the probability of inserting the token $j$ into $\tilde{Y}(\tau_{0:t-1})$ before the token at the $i$-th position.

The encoder component of the model can have any task-specific network architecture. For Machine Translation task, it can be an arbitrary sequence encoder: any combination of RNN [3, 2], CNN [12, 16] or self-attention [10, 17]. For image-to-sequence problems (e.g. Image-To-LaTeX [18]) any 2d convolutional encoder architecture from the domain of computer vision can be used [19, 20, 21].

### 3.1 Relation to prior work

The closest to our is the work by Gu et al. [22][4], who propose a decoding algorithm which supports flexible sequence generation in arbitrary orders through insertion operations.

In terms of modeling, they describe a similar transformer-based model but use a relative-position-based representation to capture generation orders. This effectively addresses the problem that absolute positional encodings are unknown before generating the whole sequence. While in our model positional encodings of most of the tokens change after each insertion operation and, therefore, decoder self-attention is re-applied at each generation step, the model by Gu et al. [22] does not need this and has better theoretical time complexity of $O(len(Y)^2)$ in contrast to our $O(len(Y)^3)$. However, in practice our decoding is on average only $50\%$ times slower than the baseline; for the details, see Section 5.2.

In terms of training objective, they use lower bound (5) with beam search over $T^*(Y)$, which is different from our lower bound (6). However, we found our lower bound to be beneficial in terms of quality and less prone to getting stuck in local optima. We will discuss this in detail in Section 5.1.

## 4 Experimental setup

We consider three sequence generation tasks: Machine Translation, Image-To-Latex and Image Captioning. For each, we now define input $X$ and output $Y$, the datasets and the task-specific encoder

we use. Decoders for all tasks are Transformers in *base* configuration [10] (either original or INTRUS) with identical hyperparameters.

**Machine Translation**    For MT, input and output are sentences in different languages. The encoder is the Transformer-base encoder [10].

Wu et al. [7] suggest that left-to-right NMT models fit better for right-branching languages (e.g., English) and right-to-left NMT models fit better for left-branching languages (e.g., Japanese). This defines the choice of language pairs for our experiments. Our experiments include: En-Ru and Ru-En WMT14; En-Ja ASPEC [23]; En-Ar, En-De and De-En IWSLT14 Machine Translation data sets. We evaluate our models on WMT2017 test set for En-Ru and Ru-En, ASPEC test set for En-Ja, concatenated IWSLT tst2010, tst2011 and tst2012 for En-De and De-En, and concatenated IWSLT tst2014 and tst2013 for En-Ar.

Sentences of all translation directions except the Japanese part of En-Ja data set are preprocessed with the Moses tokenizer [24] and segmented into subword units using BPE [25] with 32,000 merge operations. Before BPE segmentation Japanese sentences were firstly segmented into words[5].

**Image-To-Latex**    In this task, $X$ is a rendered image of LaTeX markup, $Y$ is the markup itself. We use the ImageToLatex-140K [18, 26] data set. We used the encoder CNN architecture, preprocessing pipeline and evaluation scripts by Singh [26][6].

**Image captioning**    Here $X$ is an image, $Y$ is its description in natural language.  We use MSCOCO [27], the standard Image Captioning dataset. Encoder is VGG16 [19] pretrained[7] on the ImageNet task without the last layer.

**Evaluation**    We use BLEU[8] [29] for evaluation of Machine Translation and Image-to-Latex models. For En-Ja, we measure character-level BLEU to avoid infuence on word segmentation software. The scores on MSCOCO dataset are obtained via the official evaluation script[9].

**Training details**    The models are trained until convergence with base learning rate 1.4e-3, 16,000 warm-up steps and batch size of 4,000 tokens. We vary the learning rate over the course of training according to [10] and follow their optimization technique.  We use beam search with the beam between 4 and 64 selected using the validation data for both baseline and INTRUS, although our model benefits more when using even bigger beam sizes. The pretraining phase of INTRUS is $10^5$ batches.

## 5    Results

Table 1: The results of our experiments. En-Ru, Ru-En, En-Ja, En-Ar, En-De and De-En are machine translation experiements. $*$ indicates statistical significance with $p$-value of 0.05, computed via bootstrapping [30].

| Model | En-Ru | Ru-En | En-Ja | En-Ar | En-De | De-En | Im2Latex | MSCOCO | |
|---|---|---|---|---|---|---|---|---|---|
| | BLEU | | | | | | | BLEU | CIDEr |
| Left-to-right | 31.6 | 35.3 | 47.9 | **12.0** | 28.04 | **33.17** | 89.5 | 18.0 | 56.1 |
| Right-to-left | - | - | 48.6 | 11.5 | - | - | - | - | - |
| INTRUS | **33.2**$^*$ | **36.4**$^*$ | **50.3**$^*$ | 12.2 | **28.36**$^*$ | 33.08 | **90.3**$^*$ | **25.6**$^*$ | **81.0**$^*$ |

Among all tasks, the largest improvements are for Image Captioning: 7.6 BLEU and 25.1 CIDER.

For Machine Translation, INTRUS substantially outperforms the baselines for most considered language pairs and matches the baseline for the rest. As expected, the right-to-left generation order is better than the left-to-right for translation into Japanese. However, our model significantly outperforms both baselines. For the tasks where left-to-right decoding order provides a strong inductive bias (e.g. in De-En translation task, where source and target sentences can usually be aligned without any permutations), generation in arbitrary order does not give significant improvements.

Image-To-Latex improves by 0.8 BLEU, which is reasonable difference considering the high performance of the baseline.

## 5.1 Ablation analysis

In this section, we show the superior performance of the proposed lower bound of the data log-likelihood (6) over the natural choice of (5). We also emphasize the importance of the pretraining phase for INTRUS. Specifically, we compare performance of the following models:

- **Default** — using the training procedure described in Section 2.2;
- **Argmax** — trained with the lower bound (5) (maximum is approximated with using beam search with the beam of 4; this technique matches the one used in Gu et al. [22]);
- **Left-to-right pretraining** — pretrained with the fixed left-to-right decoding order (in contrast to the uniform samples in the default setting);
- **No pretraining** — with no pretraining phase;
- **Only pretraining** — training is performed with a model-independent order, either uniform or left-to-right.

Table 2: Training strategies of INTRUS. MT task, scores on the WMT En-Ru 2012-2013 test sets.

| Training strategy | INTRUS | Argmax | Pretraining left-to-right | No pre-training | Only pretraining | | Baseline left-to-right |
|---|---|---|---|---|---|---|---|
| | | | | | uniform | left-to-right | |
| BLEU | 27.5 | 26.6 | 26.3 | 27.1 | 24.6 | 25.5 | 25.8 |

Table 2 confirms the importance of the chosen pretraining strategy for the performance of the model. In preliminary experiments, we also observed that introducing any of the two pretraining strategies increases the overall robustness of our training procedure and helps to avoid convergence to poor local optima. We attribute this to the fact that a pretrained model provides the main algorithm with a good initial exploration of the trajectory space $T^*$, while the Argmax training strategy tends to quickly converge to the current best trajectory which may not be globally optimal. This leads to poor performance and unstable results. This is the only strategy that required several consecutive runs to obtain reasonable quality, despite the fact that it starts from a good pretrained model.

## 5.2 Computational complexity

Despite its superior performance, INTRUS is more computationally expensive compared to the baseline. The main computational bottleneck in the model training is the generation of insertions required to evaluate the training objective (6). This generation procedure is inherently sequential. Thus, it is challenging to effectively parallelize it on GPU accelerators. In our experiments, training time of INTRUS is 3-4 times longer than that of the baseline. The theoretical computational complexity of the model's inference is $O(|Y|^3 k)$ compared to $O(|Y|^2 k)$ of conventional left-to-right models. However, in practice this is likely not to cause drastic decrease of the decoding speed. Figure 4 shows the decoding speed of both INTRUS and the baseline measured for machine translation task. On average, INTRUS is only $50\%$ slower because for sentences of a reasonable length it performs comparably to the baseline.

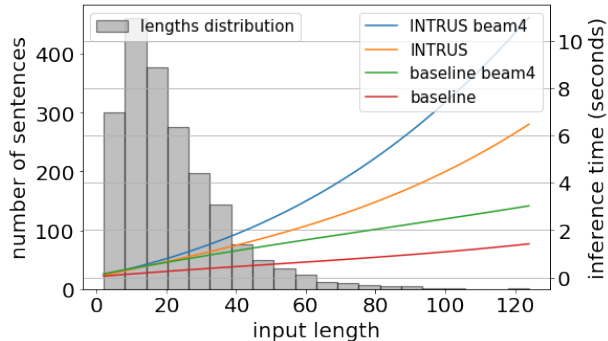

Figure 4: Inference time of INTRUS and the baseline models vs sentence length.

# 6 Analyzing learned generation orders

In this section, we analyze generation orders learned by INTRUS on the Ru-En translation task.

**Visual inspection** We noticed that the model often follows a general decoding direction that varies from sentence to sentence: left-to-right, right-to-left, middle-out, etc. (Figure 5 shows several examples[10]). When following the chosen direction, the model deviates from it for translation of certain phrases. For instance, the model tends to decode pairs of quotes and brackets together. Also we noticed that tokens which are generated first are often uninformative (e.g., punctuation, determiners, etc.). This suggests that the model has preference towards generating "easy" words first.

```
but the researchers gave them pieces of wire .      actually , it was not so funny .      the study was conducted among 900 children .
but the researchers gave them pieces of wire .      actually , it was not so funny .      the study was conducted among 900 children .
but the researchers gave them pieces of wire .      actually , it was not so funny .      the study was conducted among 900 children .
but the researchers gave them pieces of wire .      actually , it was not so funny .      the study was conducted among 900 children .
but the researchers gave them pieces of wire .      actually , it was not so funny .      the study was conducted among 900 children .
but the researchers gave them pieces of wire .      actually , it was not so funny .      the study was conducted among 900 children .
but the researchers gave them pieces of wire .      actually , it was not so funny .      the study was conducted among 900 children .
but the researchers gave them pieces of wire .      actually , it was not so funny .      the study was conducted among 900 children .
but the researchers gave them pieces of wire .      actually , it was not so funny .      the study was conducted among 900 children .
```

Figure 5: Decoding examples: left-to-right (left), right-to-left (center) and middle-out (right). Each line represents one decoding step.

**Part of speech generation order** We want to find out if the model has any preference towards generating different parts of speech in the beginning or at the end of the decoding process. For each part of speech,[11] we compute the relative index on the generation trajectory (for the baseline, it corresponds to its relative position in a sentence). Figure 6 shows that INTRUS tends to generate punctuation tokens and conjunctions early in decoding. Other parts of speech like nouns, adjectives, prepositions and adverbs are the next easiest to predict. Most often they are produced in the middle of the generation process, when some context is already established. Finally, the most difficult for the model is to insert verbs and particles.

These observations are consistent with the easy-first generation hypothesis: the early decoding steps mostly produce words which are the easiest to predict based on the input data. This is especially interesting in the context of previous work. Ford et al. [8] study the influence of token generation order on a language model quality. They developed a family of two-pass language models that depend on a partitioning of the vocabulary into a set of first-pass and second-pass tokens to generate sentences. The authors find that the most effective strategy is to generate function words in the first pass and content words in the second. While Ford et al. [8] consider three manually defined strategies, our model learned to give preference to such behavior despite not having any inductive bias to do so.

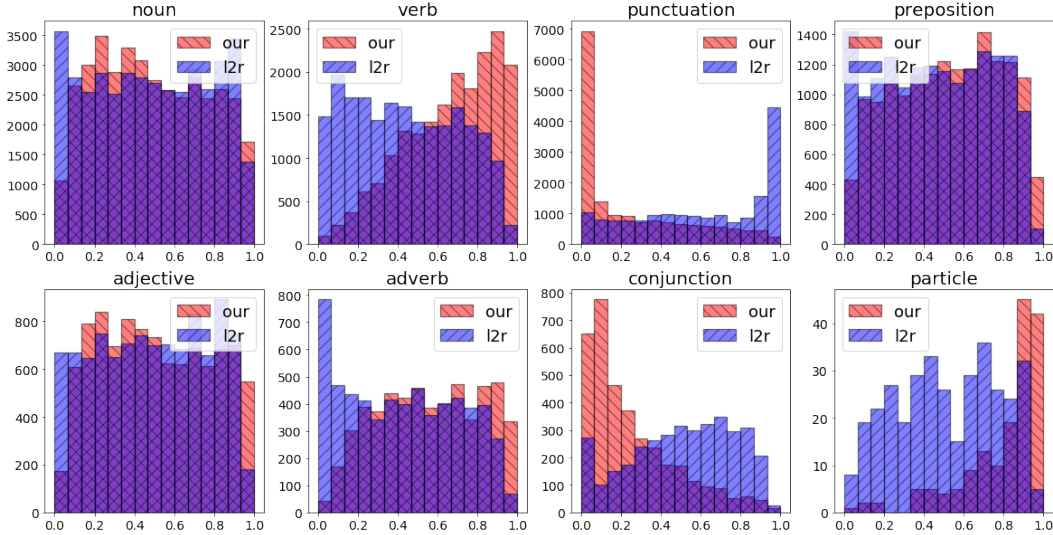

Figure 6: The distributions of the relative generation order of different parts of speech.

# 7 Related work

In Machine Translation, decoding in the right-to-left order improves performance for English-to-Japanese [32, 7]. The difference in translation quality is attributed to two main factors: Error Propagation [33] and the concept of language branching [7, 34]. In some languages (e.g. English), sentences normally start with subject/verb on the left and add more information in the rightward direction. Other languages (e.g. Japanese) have the opposite pattern.

Several works suggest to first generate the most "important" token, and then the rest of the sequence using forward and backward decoders. The two decoders start generation process from this first "important" token, which is predicted using classifiers. This approach was shown beneficial for video captioning [6] and conversational systems [35]. Other approaches to non-standard decoding include multi-pass generation models [36, 8, 37, 38] and non-autoregressive decoding [39, 38].

Several recent works proposed sequence models with arbitrary generation order. Gu et al. [22] propose a similar approach using another lower bound of the log-likelihood which, as we showed in Section 5.1, underperforms ours. They, however, achieve $O(|Y|^2)$ time complexity by utilizing a different probability parameterization along with relative position encoding. Welleck et al. [40] investigates the possibility of decoding output sequences by descending a binary insertion tree. Stern et al. [41] focuses on parallel decoding using one of several pre-specified generation orders.

# 8 Conclusion

In this work, we introduce INTRUS, a model which is able to generate sequences in any arbitrary order via iterative insertion operations. We demonstrate that our model learns convenient generation order as a by-product of its training procedure. The model outperforms left-to-right and right-to-left baselines on several tasks. We analyze learned generation orders and show that the model has a preference towards producing "easy" words at the beginning and leaving more complicated choices for later.

# Acknowledgements

The authors thank David Talbot and Yandex Machine Translation team for helpful discussions and inspiration.

## Footnotes

[1]The source code is available at https://github.com/TIXFeniks/neurips2019_intrus.

[2]To justify this transition, we translated $10^4$ sentences with a fully trained model using beam size 128 and found only 4 occasions where multiple insertion trajectories in the beam led to the same output sequence.

[3]Though this makes training more computationally expensive than the standard Transformer, this does not hurt decoding speed much: on average decoding is only 50% times slower than the baseline. We will discuss this in detail in Section 5.2.

[4]At the time of submission, this was a concurrent work.

[5]Open-source word segmentation software is available at `https://github.com/atilika/kuromoji`

[6]We used `https://github.com/untrix/im2latex`

[7]We use pretrained weights from keras applications `https://keras.io/applications/`, the same for both baseline and our model.

[8]BLEU is computed via SacreBLEU [28] script with the following parameters: BLEU+c.lc+l.[src-lang]-[dst-lang]+.1+s.exp+tok.13a+v.1.2.18

[9]Script is available at `https://github.com/tylin/coco-caption`

[10]More examples and the analysis for Image Captioning are provided in the supplementary material.

[11]To derive part of speech tags, we used CoreNLP tagger [31].

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
