[Supplementary Material · neurips_levenshteins_monster_supplementary.pdf]

# 1 MSCOCO part of speech analysis

The distributions on Figure 1 generated on MSCOCO Image Captioning dataset demonstrate that, unlike for translation task, conjunctions are not produced in the beginning of generation, as they can no longer be copied from the source, but rather depend on the currently generated sequence. Adjectives and adverbs are now mostly produced in the end of generation as a further refinement. The adjectives and adverbs are hard to generate first based on the image alone, because it is hard to tell in advance what aspects of the image the model will describe in the generated sentence.

Figure 1: The distributions of the relative generation order of different parts of speech on MSCOCO data set.

Figure 2: Learned order dependency inversion fractions for several dependency types. The full dependency taxonomy is available on universal dependencies website: https://universaldependencies.org/docs/en/dep/.

## 2  Decoding examples

```
and death is part of everyday life .
and death is part of everyday life .
and death is part of everyday life .
and death is part of everyday life .
and death is part of everyday life .
and death is part of everyday life .
and death is part of everyday life .
and death is part of everyday life .
```

Figure 3: Example of Ru-En decoding order.

```
the army faces mounting pressure on rape advocates
the army faces mounting pressure on rape advocates
the army faces mounting pressure on rape advocates
the army faces mounting pressure on rape advocates
the army faces mounting pressure on rape advocates
the army faces mounting pressure on rape advocates
the army faces mounting pressure on rape advocates
the army faces mounting pressure on rape advocates
```

Figure 4: Example of Ru-En decoding order.

```
the study was conducted among 900 children .
the study was conducted among 900 children .
the study was conducted among 900 children .
the study was conducted among 900 children .
the study was conducted among 900 children .
the study was conducted among 900 children .
the study was conducted among 900 children .
the study was conducted among 900 children .
```

Figure 5: Example of Ru-En decoding order.

```
some stories tell about honor and courage .
some stories tell about honor and courage .
some stories tell about honor and courage .
some stories tell about honor and courage .
some stories tell about honor and courage .
some stories tell about honor and courage .
some stories tell about honor and courage .
some stories tell about honor and courage .
```

Figure 6: Example of Ru-En decoding order.

```
the costs will make millions of euros .
the costs will make millions of euros .
the costs will make millions of euros .
the costs will make millions of euros .
the costs will make millions of euros .
the costs will make millions of euros .
the costs will make millions of euros .
the costs will make millions of euros .
```

Figure 7: Example of Ru-En decoding order.

```
the study was conducted among 900 children .
the study was conducted among 900 children .
the study was conducted among 900 children .
the study was conducted among 900 children .
the study was conducted among 900 children .
the study was conducted among 900 children .
the study was conducted among 900 children .
the study was conducted among 900 children .
```

Figure 8: Example of Ru-En decoding order.

```
a kitchen with a lot of stuff on the counter .
a kitchen with a lot of stuff on the counter .
a kitchen with a lot of stuff on the counter .
a kitchen with a lot of stuff on the counter .
a kitchen with a lot of stuff on the counter .
a kitchen with a lot of stuff on the counter .
a kitchen with a lot of stuff on the counter .
a kitchen with a lot of stuff on the counter .
a kitchen with a lot of stuff on the counter .
a kitchen with a lot of stuff on the counter .
a kitchen with a lot of stuff on the counter .
```

Figure 9: Example of MSCOCO decoding order.

Figure 10: Example of Ru-En decoding order.

Figure 11: Example of Ru-En decoding order.

Figure 12: Example of Ru-En decoding order.

a young girl standing in front of a large advertisement .
a young girl standing in front of a large advertisement .
a young girl standing in front of a large advertisement .
a young girl standing in front of a large advertisement .
a young girl standing in front of a large advertisement .
a young girl standing in front of a large advertisement .
a young girl standing in front of a large advertisement .
a young girl standing in front of a large advertisement .
a young girl standing in front of a large advertisement .
a young girl standing in front of a large advertisement .
a young girl standing in front of a large advertisement .

Figure 13: Example of MSCOCO decoding order.

three people are riding on the back of an elephant .
three people are riding on the back of an elephant .
three people are riding on the back of an elephant .
three people are riding on the back of an elephant .
three people are riding on the back of an elephant .
three people are riding on the back of an elephant .
three people are riding on the back of an elephant .
three people are riding on the back of an elephant .
three people are riding on the back of an elephant .
three people are riding on the back of an elephant .
three people are riding on the back of an elephant .

Figure 14: Example of MSCOCO decoding order.

a seagull that is sitting on a raft in the water .
a seagull that is sitting on a raft in the water .
a seagull that is sitting on a raft in the water .
a seagull that is sitting on a raft in the water .
a seagull that is sitting on a raft in the water .
a seagull that is sitting on a raft in the water .
a seagull that is sitting on a raft in the water .
a seagull that is sitting on a raft in the water .
a seagull that is sitting on a raft in the water .
a seagull that is sitting on a raft in the water .
a seagull that is sitting on a raft in the water .
a seagull that is sitting on a raft in the water .

Figure 15: Example of MSCOCO decoding order.

a group of traffic sitting in front of a building .
a group of traffic sitting in front of a building .
a group of traffic sitting in front of a building .
a group of traffic sitting in front of a building .
a group of traffic sitting in front of a building .
a group of traffic sitting in front of a building .
a group of traffic sitting in front of a building .
a group of traffic sitting in front of a building .
a group of traffic sitting in front of a building .
a group of traffic sitting in front of a building .
a group of traffic sitting in front of a building .

Figure 16: Example of MSCOCO decoding order.