[Reviews · NeurIPS 2019]

Reviewer 1



Updated review: The authors have indicated that they will run additional experiments and make the clarifications I requested, so I will raise my score in 7 in agreement with the other reviews leading to an "accept" consensus. However, I do note that in their rebuttal the authors describe Gu et al., Stern et al., and Welleck et al. as "concurrent work". To be totally clear, all three of those papers were posted to arxiv in early February; the NeurIPS deadline was over 3 months later and it is now 6 months after they papers appeared online. I would argue that 3 (or 6) months is long enough to provide a more direct comparison and would not consider this submission "concurrent work". I don't think this warrants rejecting the paper, but I do want to note that I disagree with the authors here and still believe that a more direct comparison is appropriate. Summary: This paper proposes a method for generating output sequences in an arbitrary order in sequence-to-sequence tasks. To do so, the model outputs an insertion location and a token at each output timestep. To avoid the combinatorial search problem which results from natural optimization of this procedure, the authors propose a lower bound which can be computed reasonably efficiently. Experimental results are given on machine translation and image-to-text tasks, and the proposed approach consistently outperforms left-to-right and right-to-left generation. Review: Overall this is an important problem to tackle, and an interesting approach. The main drawback of the method is its increased computational complexity, but if it consistently leads to better performance then the additional complexity is arguably worthwhile. The paper is overall written reasonably clearly (though I have some questions and suggestions for improvement below). Overall, I think it's a solid contribution which warrants publication. However, I have one significant criticism: It is experimentally somewhat weak. While the authors consider some common benchmarks, they reimplement only a limited set of baselines and do not compare directly to previously produced numbers. The results would be much stronger if 1) additional baseline methods were included, e.g. at the very least *all* of the related methods of Section 7, 2) this idea was applied to a stronger baseline (e.g. some SoTA seq2seq model) and improved performance was still observed, and 3) a more direct comparison to published numbers on the studied datasets was provided. As it currently stands I will give it a weak accept; I will be happy to raise my score if the authors make these improvements. Specific comments/questions: - Why not consider tasks without language output? That would help show that the gains from this method are not specific to language generation tasks. For example, you could consider music generation tasks, where a "good" generation order is not obvious. - You find that "the model has a preference towards producing 'easy' words at the beginning and leaving more complicated choices for later". Why not include this as a baseline for comparison? - Notation issue: You formulate your model as predicting \tau from the input X, a partial output ~Y(\tau_{0:t-1}), and parameters \theta. In (6) and later, you compute an expectation over \tau \sim p(\tau | X, T*(Y), \theta). But T*(Y) is defined as "the set of all trajectories leading to Y", which is different from a (single) partial output ~Y(\tau_{0:t-1}). Do you mean that you also compute an expectation over sampling from T*(Y) first, and then use that to sample \tau? - The footnote 8 on page 6 is important. You should have introduced [28] earlier and more explicitly. - In Section 3, you write "Our model does not require the attention mask preventing attention to subsequent positions. Decoder self-attention is re-applied at each decoding step because the positional encodings of most tokens change when inserting a new one into an incomplete subsequence of Y." This will make the reader wonder the extent to which your model is parallelizable. You only address this later in Section 5.2. I would suggest moving some of this discussion up earlier; it will make the presentation of the algorithm clearer.

Reviewer 2



The core training method in the paper builds closely off of Vinyals (2016), but with a substantially different model and applying it to much larger and more realistic tasks. The paper provides experiments with several different modeling choices and ablations for comparison. The experiments would be better if they had a direct comparison to the evaluation numbers in prior work such as Gu rather than their own implementation which may differ in the details. The analysis of insertion order by part of speech was interesting. The method does have some speed disadvantages, but it is not prohibitively expensive and the paper provides a clear comparison of runtime.

Reviewer 3



Originality: This work proposes a model which jointly predicts the label and its position to insert it. It is formulated as a variant of ordered set prediction, but slightly differs in that the set grows as we predict a label together with the position to insert it given the current set. The sampling based training procedure is based on the prior work but it sounds quite natural to me. Quality: The proposed approach is well-motivated and technically sound to me, in which the prediction is split into two, position and token sub-models. Experiments are well-designed and its computation complexity for inference sounds correct to me. The analysis is quite interesting in that the generation order looks intuitive and the results are analyzed further in terms of the generation order differentiated by POSs. Clarity: It is clearly written. However, it is a little bit unclear about the notion of positions, whether they are global positions, i.e., sentence positions, or local positions, i.e., positions of the currently generated set. It would be good to introduce notations so that each \tau_t is dependent on the length of the generated labels at time t. Significance: It is an interesting model and might be of interest to many researchers. However, I'd rather expect more experiments on standard data set, e.g., WMT en-de, for comparison with prior work.

[Author Response · NeurIPS 2019]

# 1 General response

We thank all reviewers for their valuable feedback and thoughtfull suggestions.

> Comparison with Gu et al., Stern et al, Welleck et al

- > Direct comparison with Gu et al.
  To the best of our knowledge, there is no official implementation for the paper by Gu et al. (no link to the code in the paper, no repository stating that it's an implementation for the paper). Thus, direct comparison using the original implementation was not possible.
  However, in Section 5.1 we compare the lower bound on the objective we use with the one of Gu et al.(2019), when applied to our model. This is a more fair comparison of the two approaches since both model architectures can be used with these lower bounds and should be compared separately.

- > Comparison with Stern et al, Welleck et al
  These works do not report significant improvements in BLEU scores against the autoregressive baselines. Stern et al.(2019) focus on parallel decoding (with the final result matching the vanilla Transformer). Welleck et al.(2019) find left-to-right models superior to their approach.

While we took into account these papers for the presentation of our work, we want to highlight that this was concurrent work.

> More experiments on standard data sets, e.g., WMT EN-DE

We agree that this is important for future research on this topic and would allow for easier comparison with our and previous work. While we can not provide these results during the author response period (due to large training time of NMT models for high-resource language pairs), we will add them should the paper get accepted.

# 21 Response to reviewer 1

> Why not consider tasks without language output? ... For example, you could consider music generation tasks, ...

Note that we consider not only natural language output, but also Image-to-Latex, where output is LaTex formulas.

We believe applying our approach to music generation task could be an interesting direction for future work.

> Notation issue: ... Do you mean that you also compute an expectation over sampling from $T^*(Y)$ first, and then use that to sample $\tau$?

By $\tau \sim p(\tau|X, T^*(Y), \theta)$ we actually mean $\tau \sim p(\tau|X, \tau \in T^*(Y), \theta)$. Thus, we sample the trajectory from our model, but only allow the correct insertion operations (i.e. the ones that lead to producing $Y$).

We will make it more clear and expand the discussion.

> The footnote 8 on page 6 is important. You should have introduced [28] earlier and more explicitly.

> In Section 3, you write ... I would suggest moving some of this discussion up earlier; it will make the presentation of the algorithm clearer.

We agree this would improve the presentation. We will revise the paper accordingly.

# 34 Response to reviewer 3

> how to decide the output length

We describe this in Section 2.2. Specifically, we introduce the EOS element, which indicates the end of generation process. The model can produce this element instead of a pair of position and a token.

[Meta-Review · NeurIPS 2019]

The paper deals with the interesting approach of generating a sequence given a token graph and demonstrate its effectiveness on several practical setting.